# Modifications of Titin Contribute to the Progression of Cardiomyopathy and Represent a Therapeutic Target for Treatment of Heart Failure

**DOI:** 10.3390/jcm9092770

**Published:** 2020-08-26

**Authors:** Charles Tharp, Luisa Mestroni, Matthew Taylor

**Affiliations:** Adult Medical Genetics Program, Cardiovascular Institute, University of Colorado Anschutz Medical Campus, CO 80045, USA; charles.tharp@cuanschutz.edu (C.T.); luisa.mestroni@cuanschutz.edu (L.M.)

**Keywords:** titin, RNA binding motif protein 20 (RBM20), sarcomere, systolic dysfunction, diastolic dysfunction, dilated cardiomyopathy, phosphorylation, non-sense mRNA decay, mammalian target of rapamycin (mTOR) complex-1

## Abstract

Titin is the largest human protein and an essential component of the cardiac sarcomere. With multiple immunoglobulin(Ig)-like domains that serve as molecular springs, titin contributes significantly to the passive tension, systolic function, and diastolic function of the heart. Mutations leading to early termination of titin are the most common genetic cause of dilated cardiomyopathy. Modifications of titin, which change protein length, and relative stiffness affect resting tension of the ventricle and are associated with acquired forms of heart failure. Transcriptional and post-translational changes that increase titin’s length and extensibility, making the sarcomere longer and softer, are associated with systolic dysfunction and left ventricular dilation. Modifications of titin that decrease its length and extensibility, making the sarcomere shorter and stiffer, are associated with diastolic dysfunction in animal models. There has been significant progress in understanding the mechanisms by which titin is modified. As molecular pathways that modify titin’s mechanical properties are elucidated, they represent therapeutic targets for treatment of both systolic and diastolic dysfunction. In this article, we review titin’s contribution to normal cardiac physiology, the pathophysiology of titin truncation variations leading to dilated cardiomyopathy, and transcriptional and post-translational modifications of titin. Emphasis is on how modification of titin can be utilized as a therapeutic target for treatment of heart failure.

## 1. Titin Is an Essential Molecule within the Sarcomere Where It Contributes to Cardiac Function

### 1.1. Titin Contains Four Structural Domains That Impart Important Molecular Properties to the Sarcomere during Cardiac Function

The essential function of the heart is generation of circulatory blood flow to organs and tissues. This circulatory flow or cardiac output is created by coordinated contraction and relaxation of the heart muscle, termed systole and diastole, respectively. During systole, the heart muscle contracts and ejects blood from the heart delivering oxygen and vital nutrients to the body. During diastole, the heart muscle relaxes allowing blood to fill the ventricles in preparation for systole. Dysfunction of systole and diastole can lead to the development of heart failure. Heart failure is a clinical syndrome where dysfunction affecting systole, diastole or both leads to significant symptoms and mortality. Heart failure is one of the most morbid health conditions affecting an estimated 5.1 million people in the United States and 23 million people worldwide [1,2]. As a basic principle, heart failure results from dysfunction of the cardiomyocyte and sarcomeric structures. Cardiomyocytes are specialized muscle cells that contain sarcomeres which are the fundamental sub-cellular structure that convert chemical energy to muscular contraction. The sarcomere is a complex protein structure with rigid vertical M-lines and Z-lines that anchor overlapping longitudinal filaments. During cardiac contraction, the myosin heads pull actin filaments in one direction bringing the vertical M-line and Z-lines together resulting in contraction and mechanical force generation [3].

One of the most important proteins that compose the sarcomere is titin. Titin is a filamentous protein anchored in the Z-line and extending to the M-line thereby providing structural support to the sarcomere [4,5]. Titin is the largest known protein with a measured length of up to 1.2 μM and comprised of 27,000–33,000 amino acids with a molecular mass of 3800 kDa [6,7]. Titin is encoded by the titin gene *TTN*, which is 294 kb and contains 364 exons [8]. Although titin has many complex molecular properties, its mechanical role supporting the sarcomere and providing passive tension and force modulation during contraction and relaxation has been the most well described and are the focus of this paper [5,9]. The molecular properties of titin as it contributes to normal function are imparted by the four structural domains (Figure 1). The N-terminal domain embeds and anchors it to the Z-disk, while the C-terminal domain similarly anchors titin to the M-band [4,5,10]. Titin filaments in the Z-disk and M-line are embedded to domains of adjacent titin proteins from the opposing sarcomere to form overlapping connections that create a continuous filamentous connection spanning the myofibril [11]. The I-band is adjacent to the Z-disk and is composed of repetitive immunoglobulin-like domains, a region rich in proline (P), glutamate (E), valine (V), and lysine (K) termed the PEVK region, as well as the N2B element [5,11]. The I-band is highly elastic and provides a spring-like function that imparts passive tension during relaxation, elongates during contraction, and returns the sarcomere to its previous length after contraction [12,13]. The A-band serves as a rigid region that binds to the thick filament and interacts with myosin to translate force during contraction to the sarcomere [14].

### 1.2. Titin’s Structure Is Essential to Normal Cardiac Function, Passive Tension, and Length Dependent Activation

The structure of titin domains provides essential functions to the cardiomyocyte and overall physiology of the heart. The elasticity of titin that is derived from the spring-like I-band is a major contributor to the length dependent activation described in the Frank-Starling effect. In normal cardiac function, increased blood volume in the ventricle during diastole (measured by left ventricular end diastolic pressure) leads to increased force of contraction during systole (measured by stroke volume; see Figure 2). The Frank-Starling effect is an essential principal of cardiac function that allows the heart to compensate for physiological changes and increased demands for cardiac output [15]. While there are many factors that influence the relationship between diastolic filling and stroke volume, titin’s ability to stretch is an essential contributor. As the sarcomere expands in diastole, the springy Ig-like domains of titin’s I-band elongate, thereby increasing tension that is released as restorative force during systole. This increased diastolic tension also modulates actomyosin interactions to increase the force of contraction during systole [16,17]. Because titin spans the entirety of the sarcomere, its length and elasticity are major contributors to the passive tension of the ventricle including diastolic tension and diastolic volume [18]. In addition to the mechanical support titin provides to the sarcomere, stretching of titin leads to cellular signaling activation that promotes myocyte growth and contributes to chronic changes of the myocardium [15].

### 1.3. Abnormalities in Titin Contribute to Systolic and Diastolic Heart Failure

The essential function of titin in the sarcomere and cardiac function is illustrated by the significant cardiac phenotypes in patients who have *TTN* mutations. In addition, patients who develop heart failure related to other causes have significant changes to titin at the transcriptional and post translational level. Cardiomyopathy and heart failure are categorized clinically into systolic dysfunction and diastolic dysfunction. In systolic dysfunction, there is decreased contraction of the heart leading to decreased stroke volume and elevated cardiac pressures. Systolic dysfunction is categorized into dilated cardiomyopathy and hypertrophic cardiomyopathy. In dilated cardiomyopathy, dysfunction of the cardiomyocytes during contraction leads to remodeling of the myocardium in an eccentric pattern which causes dilation of the ventricle. In hypertrophic cardiomyopathy, changes in myocardial function result in concentric hypertrophy and thickening of the ventricle. Restrictive cardiomyopathy is caused by progression of diastolic dysfunction where the contractile force of the heart is preserved; however, due to abnormal relaxation of the heart during diastole, the ventricle cannot fill appropriately resulting in decreased cardiac function. In both systolic and diastolic dysfunction, there are elevated cardiac filling pressures leading to the clinical presentation of heart failure with development of peripheral and pulmonary edema, dyspnea, and fatigue.

We will review how mutations in *TTN* contribute to cardiomyopathy. We will discuss how transcriptional changes and regulation of titin affect cardiac physiology. We will describe post-translational modifications of titin that occur with heart failure. In covering these topics, there will be a translational focus on associating molecular pathways with clinical phenotypes and how these pathways may lead to novel therapeutic targets for treatment of cardiomyopathy.

## 2. Truncation Mutations in *TTN* Cause Dilated Cardiomyopathy (DCM)

### 2.1. TTN Truncation Variations in Highly Constitutively Expressed Exons Cause DCM

*TTN* mutations are the most common cause of genetic DCM due in part to *TTN’s* large gene size as well as its essential role in the sarcomere and cardiomyocyte function. There are numerous point mutations of *TTN* leading to missense changes in single amino acids that are associated with diverse phenotypes including several types of cardiomyopathies [19,20]. However, pathogenic *TTN* mutations most commonly result from nonsense mutations leading to frameshifts and incorporation of early stop codons predicting a truncated, shortened titin protein, termed titin truncation variations (TTNtv) [21]. Interestingly, TTNtv, mostly restricted to the I-band, are found in 2–3% of the general population [13,22,23] and there is substantial interest in determining the associated impact of TTNtvs in ostensibly healthy persons. While phenotypes of TTNtvs can be diverse and may present with restrictive cardiomyopathy, they are mostly associated with DCM that presents later in life [24,25]. There is a relationship between mutation location and pathogenicity of TTNtv phenotype. Given the size of *TTN*, there are many alternative splicing events during transcription and hundreds of unique isoforms expressed in cardiac tissue. The key predictor of pathogenicity of TTNtv is whether the mutated exon is expressed in a high proportion of the total cardiac-expressed isoforms. In a population-based study with genome sequencing, RNA sequencing, and cardiac phenotyping, pathogenicity of a given TTNtv was correlated with how frequently the mutated exon is expressed. This was assigned a value, proportion spliced in (PSI) from 0 to 1, with a PSI > 0.9 imparting a 93% probability of pathogenicity [23]. The association of PSI and phenotype is further supported by pathogenicity data from the Clinvar (https://www.ncbi.nlm.nih.gov/clinvar/) database where exon locations of high PSI mutations are correlated with known pathogenic mutations [17]. This is an essential finding because *TTN* mutations are common, therefore predicting whether a given cardiac phenotype is attributable to a TTNtv mutation or an alternative cause is helpful for clinical prognostication. The PSI score as a predictor of pathogenicity fits with previous observations that A-band mutations are more pathogenic, as A-band exons are more constitutively expressed. As a corollary, Z-disk mutations are less pathogenic, likely to be due to lower constitutive expression of Z-disk exons. In addition, there is a common TTN isoform, cronos, that has an alternative downstream translational start site that can compensate for upstream truncation mutations [26]. Recently there have examples of TTNtv with high PSI without cardiac phenotypes. There is evidence that in these circumstances, exons with TTNtv can be translated either through utilization of internal ribosomal entry site or stop codon suppression [27].

### 2.2. TTNtv Cause DCM through Haploinsufficiency, Increased Metabolic Stress, and Activation of the mTOR Signalling Cascade

The exact mechanism for how TTNtv lead to a cardiac phenotype has not been clearly demonstrated. There are studies that suggest specific mutations cause direct damage to the sarcomere and cardiomyocyte. This was observed in a cell culture experiment using human induced pluripotent stem-cell derived cardiomyocytes (hIPSC-CM) isolated from a symptomatic patient with an A-band TTNtv. In the derived cardiomyocytes, a shortened titin protein was isolated and associated with disorganized sarcomere formation and impaired contractility [28]. Similarly, hiPSC-CM derived from a patient with unique A-band TTNtv demonstrated abnormal sarcomerogenesis and cardiomyocyte force generation due to loss of a β-cardiac myosin binding site on the mutated titin protein [29]. Despite these findings in cell culture experiments, abnormal titin proteins leading to direct sarcomere damage in a “poison peptide” model have not been shown in human studies. Biopsied cardiomyocytes from symptomatic patients with TTNtv did not demonstrate similar alterations in maximum force in myofibril contraction studies [30].

An alternative to the dominant negative model of TTNtv is a model of haploinsufficiency. In this more likely explanation for pathogenicity, TTNtv do not directly damage the sarcomere but instead lead to increased metabolic stress that has chronic effects on cardiomyocyte and cardiac function. Metabolic stress is generated by abnormal mRNA transcripts from the truncation variant leading to increased non-sense mRNA decay (NMD). This leads to long term compensatory changes and development of a common DCM phenotype that is independent of *TTN* mutation location [31,32,33] (Figure 3). Rat models with A-band and I-band TTNtv do not alter the amount of TTN expressed, but there is increased NMD and a shift in cardiac metabolism to preference branched chain amino acids and glycolytic intermediates instead of fatty acids that are typically utilized in healthy cardiomyocytes [34,35]. Increased metabolic stress associated with NMD activates mammalian target of rapamycin (mTOR) complex-1 signaling resulting in pathologic protein synthesis and autophagy leading to cardiac phenotypes [31,36]. This was demonstrated in a rat model with TTNtv where there was elevated mTOR complex activation and decreased antiphagocytic degradation in cells. This led to increased reactive oxygen species and mitochondrial dysfunction. Interestingly, treatment of rats with the mTOR inhibitor, rapamycin, rescued the cardiac phenotype [31]. Activation of the mTOR complex appears to be a common downstream pathway of heart failure as its activation is associated with other causes of DCM [37,38]. Further elucidation of these pathways is required to better understand how TTNtv lead to DCM phenotypes which will provide additional opportunities to develop therapies. Animal models are essential for understanding molecular pathways, and several have been utilized to study TTNtv including a mouse model with an A-band truncation mutation, and zebrafish models that have several different truncation variations [39,40,41,42]. A review of these important models has been recently carried out [43].

### 2.3. TTNtv Are Associated with Late Presentation Of DCM, Are More Pathogenic in European Ancestry, and Are Associated with Early Atrial Arrhythmias

The mechanism of haploinsufficiency leading to abnormal metabolism and increased stress is in keeping with the described human cardiac phenotypes of TTNtv. Mutations of *TTN* are the most common cause of genetic cardiomyopathy and account for 20–25% of genetic DCM [11]. TTNtv are associated with a later presentation (47.9 years) and greater longevity (70.4 years) compared to other genetic and non-genetic causes of cardiomyopathy [24,44]. While patients with TTNtv often present later in life, there is evidence that carriers have pre-symptomatic cardiac dysfunction with eccentric cardiac remodeling as detected by MRI [35]. In a study of 71,000 patients who have been genotyped and undergone cardiac phenotyping, pathogenic TTNtv defined as PSI > 0.9 imparted an increased risk of reduced systolic function regardless of symptoms. In addition, patients with pathologic TTNtv had high risk of developing DCM compared to controls. Interestingly, risk of developing DCM with pathologic TTNtv was associated with ethnicity. Patients with European ancestry and TTNtv had increased odds of developing DCM compared to healthy controls (odds ratio 18.7). Patients with African ancestry and TTNtv had odds of developing DCM that were nearer to the general population (odds ratio 1.8) [45]. While TTNtv have not been closely associated with life threatening ventricular arrhythmias, they are associated with development of atrial arrhythmias including atrial fibrillation. In a case control study of a cohort of 2781 who developed atrial fibrillation before age 66, TTNtv were found in 2.1% of patients compared to 1.1% of controls (odds ratio 1.76). In patients who develop very early atrial fibrillation before age 30, TTNtv were found in 6.5% of patients (odds ratio 5.94) suggesting TTNtv impart increased risk of developing very early atrial fibrillation [46].

### 2.4. TTNtv Are Associated with Worse Outcomes When Combined with Additional Cardiac Risk Factors

Although TTNtv are associated with late development of DCM, when combined with additional risk factors for cardiomyopathy they are associated with earlier presentations and worse outcomes. This suggests a two hit hypothesis for development of TTNtv cardiomyopathy where *TTN* haploinsufficiency increases metabolic stress and when combined with additional risks such as age, toxins, pregnancy or acquired disease can lead to development of DCM [47]. In recent case reports of patients who developed anthracycline associated chemotherapy-induced cardiomyopathy (CCM) and underwent genetic studies, TTNtv were discovered [48]. In a larger study of 213 patients with CCM, TTNtv were present in 7.5% of patients compared to 1.1% of controls. In addition, patients with CCM and TTNtv had more severe cardiac dysfunction and increased risk of atrial fibrillation compared to patients with CCM without a TTNtv [49]. These data suggest that TTNtv when combined with exposure to cardiotoxic chemotherapy increases risk for development of cardiomyopathy. Similar evidence is seen in alcohol cardiomyopathy (ACM), which is a toxic cardiomyopathy related to excessive alcohol consumption. In 141 patient with ACM screened for genetic causes, TTNtv were present in 9.9% of ACM patients compared to 0.7% in controls [50]. Using a multivariate analysis comparing patients with TTNtv versus patients without TTNtv who consume excessive alcohol but do not have clinical ACM, patients with TTNtv had an 8.7% reduction in left ventricular ejection fraction compared to patients without TTNtv [50]. In addition, TTNtv have been associated with increased risk of peripartum cardiomyopathy. In 172 women with peripartum cardiomyopathy, TTNtv were discovered in 65% of patients compared to 4.7% in healthy controls [51]. These results suggest that genetic testing to identify TTNtv in patients with cardiomyopathies of alternative etiologies may be beneficial for prognostication and family screening.

### 2.5. Inhibition of the mTOR Pathway and Antisense Oligo Nucleotide Mediated Exon Skipping Are Potential Therapies for TTNtv Related DCM

Identification of TTNtv in DCM is helpful for prognosis; however, as targeted therapies for TTNtv become available, this may also change clinical management (Table 1).Given the association of TTNtv inducing upregulation of the mTOR complex, mTOR inhibitors such as rapamycin may be reasonable to consider for therapeutic trials of TTNtv DCM [52]. There have also been promising results in correcting the frameshift and early termination in TTNtv using antisense oligonucleotide (AON) mediated exon skipping. In this strategy, single stranded oligonucleotides are designed to bind pre mRNA either at the intron-exon border of the mutated exon, or to block exon splicing motifs. The goal is to skip the mutated exon that contains the frameshift mutation via alternative splicing and prevent early termination of titin during translation. This strategy has been successfully utilized in other genetic conditions such as Duchenne Muscular dystrophy with FDA approval of an AON molecule [53]. The advantage of this approach is that separate AONs can be designed for each exon containing a TTNtv allowing a targeted approach to therapy. Feasibility of this approach has been demonstrated in a mouse model with a TTNtv where AON exon skipping resulted in rescue of the DCM phenotype [54]. AON mediated exon skipping has also been tested in hiPSC-CMs derived from a patient with TTNtv, where treatment of cardiomyocytes with AON rescued defective myofibril assembly [55]. AON mediated exon skipping represents a viable treatment strategy for patients with TTNtv with DCM that is actively being pursued.

## 3. Titin Transcriptional Modifications Are Associated with Development of Cardiomyopathy and Represent a Therapeutic Target for Inherited and Acquired Heart Failure

### 3.1. Transcriptional Changes of Titin Isosforms Alter Passive Tension and Myocardial Stiffness in Heart Failure

Due to the critical role of titin in the sarcomere and cardiomyocyte, modifications and changes to titin are seen in many forms of heart failure regardless of the etiology. Specifically, transcriptional regulation of *TTN* and selection of isoforms contribute to adaptation to cardiac changes as well as maladaptation and cardiac dysfunction. Understanding and altering transcriptional regulation of titin represents a therapeutic target for treating systolic and diastolic heart failure. Titin has six named major transcriptional isoforms that impart different structural properties based on their size and extensibility [56]. Several of these isoforms are expressed only in neonatal tissue or in skeletal muscle. The two predominant isoforms in adult cardiac tissue are N2B and N2BA [57]. These isoforms differ in the I-band domain where the N2B isoform contains a short PEVK region and few Ig domains, making it relatively short and stiff. The N2BA isoform has a longer PEVK region and more spring-like Ig domains making it relatively long and soft [56] (Figure 4). The ratio of titin isoform expression in cardiac tissue is correlated with morphology and global cardiac function. In normal hearts, the ratio of N2BA:N2B is typically 70:30 [58]. In heart tissue from patients with end stage DCM, the predominant titin isoform is the longer, softer N2BA. This correlated with decreased passive tension measured on left ventricular muscle strips and in cardio-myofibrils with estimated decreased stiffness of 10% [18,58]. The predominance of the N2BA isoform in patients with DCM also correlated with echocardiographic findings with increased end diastolic volume: pressure ratio suggesting lower global myocardial stiffness [18]. These results suggest that as heart failure progresses there is a maladaptive response leading to selection of the longer, softer N2BA isoform that decreases sarcomere passive stiffness leading to increased end diastolic volume that correlates with ventricular dilation (Figure 4).

### 3.2. RBM20 Is a Key Regulator of Titin Isoform Preference

Selecting isoform expression of titin in cardiac tissue represents a potential therapeutic target in treating both systolic and diastolic dysfunction. In systolic dysfunction including DCM, transcriptionally favoring the shorter, stiffer N2B isoform may increase resting tension, decrease end diastolic volume and improve systolic function. Conversely for diastolic dysfunction where there is impaired cardiac relaxation, favoring the longer, softer N2BA isoform may decrease passive tension and improve diastolic filling. One key regulator of transcriptional selection of titin isoforms is RNA binding motif protein 20 (RBM20) [60]. RBM20 is an RNA splicing factor that participates in the spliceosome during maturation of mRNA [57,61]. RBM20 binds introns near splice sites and adjacent to U1 and U2 small nuclear ribonucleoprotein (snRNP) binding sites to regulate transcription [62]. The exact mechanism by which RBM20 regulates titin isoform selection has not been demonstrated; however, RBM20 clearly plays an essential role. Patients with RBM20 mutations develop severe DCM and are at risk of sudden cardiac death [61,63,64,65]. Sudden cardiac death associated with RBM20 mutations are related to increased arrhythmias with this mutation that are likely caused by abnormal calcium handling and increased calcium release from the sarcoplasmic reticulum [66]. In addition, RBM20 mutations have been associated with left ventricular non-compaction [67]. Humans with RBM20 mutations have a marked increase in the N2BA isoform [60]. This is also demonstrated in animal models including a rat knockout of RBM20 [60]. Further animal models for RBM20 and its targets have been described but are not as well characterized [43]. The preference for the longer, softer N2BA isoform in RBM20 mutants results in decreased active and passive tension of cardiomyocytes contributing to dilation of the ventricle [68]. Upregulation of RBM20 to preference the shorter, stiffer N2B isoform may be appealing as a treatment for patients with DCM or systolic dysfunction. To date there has been no clear mechanism or small molecule that will increase RBM20 activity. Recent studies in rats have shown that RBM20 can be upregulated by thyroid hormone-triiodothyronine (T3) resulting in preference of the N2B isoform suggesting that modification of the thyroid pathway may be beneficial in treatment of systolic heart failure [69]. Additionally, in cultured rat cells, administration of insulin activates the mTOR kinase axis and RBM20 to preference the N2B isoform [70]. Further understanding of the thyroid and insulin pathways and how they specifically affect RBM20 and titin isoform selection is needed to identify novel therapeutics for systolic dysfunction and DCM.

### 3.3. Modifications of Titin Isoforms Represent a Therapeutic Target for Treatment of Heart Failure

Downregulation of RBM20 favors the longer, softer N2BA isoform which could improve diastolic dysfunction and serve as a therapy for heart failure with preserved ejection fraction (HFpEF). In animal models, reduction of RBM20 improved diastolic dysfunction as measured by left ventricular wall thickness, echocardiographic markers, and hemodynamic markers including end diastolic pressure [71,72]. A novel approach has been developed to complete high throughput screening of small molecules to identify compounds that decrease RBM20 levels. The initial trial identified cardenolides, which include digoxin and digitoxin, as small molecules that can effectively reduce RBM20 levels and preference N2BA isoform expression [73]. Further studies in animal models are needed to determine if cardenolides represent plausible therapies for HFpEF via reduction of RBM20 and preference for the longer softer N2BA isoform. In addition to RBM20, other molecules are likely to be capable of regulating titin isoform expression and may represent alternative therapeutic targets. Another molecule that can modify titin transcriptionally is RNA binding motif protein 24 (RBM24). RBM24 is expressed in mammalian hearts and when knocked out in mice causes DCM. In one study, RBM24 knockout mice had a disorganized sarcomere and altered titin isoform expression: however, N2BA:N2B isoform expression was not determined [74]. Further studies of RBM24 are necessary including determination of how specifically titin isoform expression is altered by RBM24. In addition to proteins involved directly in transcription, micro RNAs are capable of transcriptional regulation by binding to mRNA and targeting it for degradation. In a mouse model carrying a TTNtv that develops DCM, it was demonstrated that the micro mRNA miR-208b is significantly over-expressed. Over-expression of miR-208b was also found in patients with DCM not related to TTNtv. When miR-208b was inhibited, mice did not develop a DCM phenotype suggesting that miR-208b plays a role in development of DCM possibly through transcriptional regulation of titin, although the exact mechanism for miR-208b has not been demonstrated [75].

## 4. Post-Translational Modifications of Titin Affect Myocardial Physiology and Can Be Modified to Treat Heart Failure

### 4.1. Phosphorylation of the PEVK Element of Titin Increases Cardiomyocyte Passive Tension

In addition to genetic and transcriptional modifications, post-translational modifications of titin also play an important role in cardiac physiology and changes associated with heart failure. The most well described post-translational modification of titin is phosphorylation and dephosphorylation [76]. Based on proteomic analysis, there are hundreds of predicted phosphorylation sites on titin [77,78,79]. Only a few of these potential phosphosites have been studied and demonstrated to impart functional changes on titin with a majority of the studied sites located in the spring-like I-band domain [17]. Phosphorylation within different regions of the I-band have different effects on the functional properties of titin. The PEVK element has a predominantly negative charge, therefore phosphorylation of this element with a positively charged phosphate group increases electrostatic attraction making the extensible region more rigid [76]. Treatment of human cardiomyocytes with the protein kinase C alpha (PKCα) has been shown to cause phosphorylation of the PEVK element and increase passive tension [80]. In a study of cardiomyocytes isolated by biopsy from patients with mixed systolic and diastolic heart failure, there was greater activation of PKCα within the cardiomyocytes compared to control samples. This increase in PKCα activity also correlated with increased passive tension studied on skinned myocardial fibers [81]. PKCα is activated by excess catecholamines or hypertrophic signaling cascades and may represent a terminal pathway in heart failure that leads to increased myofilament stiffness. Activation of PKCα and the resultant increase in passive tension of cardiomyocytes may be a compensatory change to increase passive stiffness in patients with DCM in order to improve cardiac function. PKCα mediated increased passive stiffness may also represent a dysregulated pathway that leads to pathologic passive tension and may result in diastolic dysfunction [82,83]. Additional studies comparing PKCα activation and passive tension between cardiomyocytes isolated from patients with DCM and patients with diastolic dysfunction may be helpful to better understand the molecular pathways involving PKCα and heart failure. It should be considered that modification of this pathway represents a viable therapeutic target for treatment of heart failure when it is more clearly understood.

### 4.2. Phosphorylation of the N2B Unique Sequence (N2Bus) Element of Titin Decreases Cardiomyocyte Passive Tension

Functionally significant phosphosites have also been discovered within the I-band in the N2Bus. This is a positively charged region, therefore phosphorylation increases repulsion forces leading to elongation and softening of titin [84]. Kinases capable of phosphorylating this element include protein kinase A (PKA), extracellular signal-regulated kinase 2 (ERK2), protein kinase G (PKG), and calcium/calmodulin-dependent protein kinase II delta (CaMKIIδ) [17]. Dephosphorylation of the N2Bus has been demonstrated by protein phosphatase 5 (PP5) [85].

### 4.3. Diabetes Contributes to Diastolic Dysfunction via Abnormal Phosphorylation of Titin, Which Can Be Reversed by Treatment with Metformin, and Neuregulin-1 (NRG-1)

Modification of titin’s stiffness via phosphorylation and dephosphorylation represents a way to modify global cardiac function. There has been clinical association between development of diastolic dysfunction and diabetes. The exact mechanism for how diabetes contributes to diastolic dysfunction is not known, but there are data suggesting that it is related to abnormal protein phosphorylation. One common therapy for diabetes that has been associated with improvement in diastolic dysfunction is metformin. It was demonstrated in a mouse model of diastolic dysfunction that metformin increases activation of PKG leading to phosphorylation of the N2Bus element and reduced passive stiffness of the sarcomere resulting in improvement of diastolic dysfunction [86]. In another study, cardiomyocytes isolated from patients with diabetes who also had diastolic dysfunction demonstrated increased passive tension compared to controls. Concurrent with the mechanical changes, the cardiomyocytes from patients with diastolic dysfunction also showed changes to the phosphorylation of titin. This included increased activation of PKCα and phosphorylation in the PEVK element, as well as reduced activation of PKG and decreased phosphorylation of the N2Bus element. These changes in phosphorylation are both predicted to increase passive tension as was observed [87]. When metformin or insulin were applied to the cells in culture, there was increased activity of ERK2 leading to increased phosphorylation of the N2Bus element which reduces passive tension; however, this was partially counteracted by increased phosphorylation of the PEVK element by PKCα. Similar abnormalities in phosphorylation were seen in a diastolic dysfunction mouse model, and the abnormal phosphorylation was reversed by treating the animals with neuregulin-1 (NRG-1). NRG-1 increased phosphorylation of the N2Bus element via ERK2 and decreased phosphorylation of the PEVK element via PKCα. The animals showed improvement in their diastolic dysfunction as measured by reduced end-diastolic pressure [87]. This study suggests that NRG-1 can reduce passive stiffness of cardiomyocytes by specifically altering phosphorylation of titin and can improve diastolic dysfunction.

### 4.4. Activation of Cyclic Guanosine Monophosphate (cGMP) and PKG Decreases Titin Passive Tension and Represents a Viable Therapy for Treatment of Heart Failure

In addition to metformin and NRG-1, sildenafil has been proposed as a therapy for diastolic dysfunction due to its effects on phosphorylation of titin. Sildenafil inhibits phosphodiesterase-5 leading to increased nitric oxide (NO) and increased cGMP signaling. cGMP is known to activate PKG, which phosphorylates the N2Bus element of titin resulting in softening of titin and decreased myocardial stiffness [88]. In a dog model of diastolic dysfunction, treatment with sildenafil increased phosphorylation of the N2Bus element via increased activation of cGMP resulting in improved diastolic function compared to untreated controls [89]. Based on these results, sildenafil was studied in patients with diastolic dysfunction to determine whether cGMP dependent PKG phosphorylation of titin can improve clinical outcomes. Unfortunately, in this placebo controlled, randomized trial, treatment with sildenafil did not significantly improve clinical, laboratory, or echocardiographic outcomes in patients [90]. Interestingly patients who received sildenafil showed no significant increase in plasma cGMP levels suggesting that the medication did not have the desired therapeutic effect on altering PKG dependent phosphorylation. Activating cGMP to improve heart failure has been pursued further with novel therapies including vericiguat which is an oral soluble guanylate cyclase stimulator that generates cGMP. This compound was studied in patients with systolic dysfunction in a high quality randomized clinical trial. Compared to placebo, treatment with vericiguat reduced hospitalizations and cardiovascular death among patients with systolic dysfunction [91]. This represents a significant breakthrough for clinical management of heart failure with a medication that utilizes a novel mechanism of action. Although increasing levels of cGMP is likely to have many effects for improving heart failure, the effects on phosphorylation of the titin N2Bus element is likely to be significant. Further study is warranted to determine mechanistically how vericiguat alters cardiomyocytes and how it affects titin. Based on the mechanism of action, vericiguat is an appealing medication to treat diastolic dysfunction due to activation of PKG, phosphorylation of the TTN N2Bus element and softening of the titin protein. Given the clinical burden of diastolic dysfunction, and the lack of effective therapeutic medications, vericiguat should be studied in patients with diastolic dysfunction. Indeed, there are at least two clinical trials of vericiguat ongoing in patients with diastolic dysfunction [92,93].

Post-translational modifications of titin including phosphorylation and dephosphorylation of specific elements contribute to development of heart failure and represent targets for therapies to treat heart failure. Further study of post-translational modifications is needed to better understand and treat heart failure. In addition to phosphorylation, there are likely to be many other post-translational modifications of titin that exist and should be studied.

## 5. Conclusions

Heart failure is a common clinical syndrome often caused by abnormalities within sarcomeric proteins including the giant, spring like protein titin. The structure of titin, specifically Ig-like domains, impart important physiologic characteristics to cardiac tissues including length dependent activation and passive tension of the ventricle. Truncation mutations of titin are the most common genetic causes of dilated cardiomyopathy and often lead to a phenotype that presents later in life and is associated with atrial arrhythmias. The pathophysiology of TTNtv leading to DCM is likely related to increased intracellular stress associated with increased nonsense mRNA decay leading to activation of the mTOR signaling cascade which results in abnormalities in cellular metabolism and ultimate dysfunction of the cardiomyocyte. This increased metabolic stress is likely to predispose patients with TTNtv to development of DCM when combined with other cardiac risk factors such as age, chemotherapy, alcohol use, and pregnancy. There are promising therapies for treating TTNtv including mTOR inhibitors and oligonucleotides that lead to alternative splicing that exclude mutated exons (Table 1).

Transcriptional modifications of titin are associated with both systolic and diastolic dysfunction and therefore titin represents a viable therapeutic target for treatment of cardiomyopathy. Patients with systolic heart failure have increased expression of the N2BA titin transcriptional variant that encodes a longer and softer titin protein, which correlates with decreased passive tension and increased left ventricular end diastolic volume. Therefore, transcriptional modification of titin to preference the shorter, stiffer N2B isoform may improve systolic heart failure. Conversely, in patients with abnormal ventricular relaxation, favoring the longer, softer titin isoform may improve diastolic filling and treat diastolic dysfunction. Although we do not yet have therapies to modulate transcription of titin, RBM20 is a key transcriptional regulator and therapeutic target (Figure 5).

Post-translational modifications of titin by phosphorylation of the spring-like I-band are capable of changing the length and stiffness of titin and are correlated with cardiomyopathy. Increased phosphorylation of the PEVK element increases titin passive tension, whereas phosphorylation of the N2Bus element decreases passive tension. Many molecules and signaling pathways affect phosphorylation of these regions and further understanding of their control may provide an opportunity to selectively modify titin and improve both systolic and diastolic heart failure (Table 1, Figure 5).

## Figures and Tables

**Figure 1 jcm-09-02770-f001:**
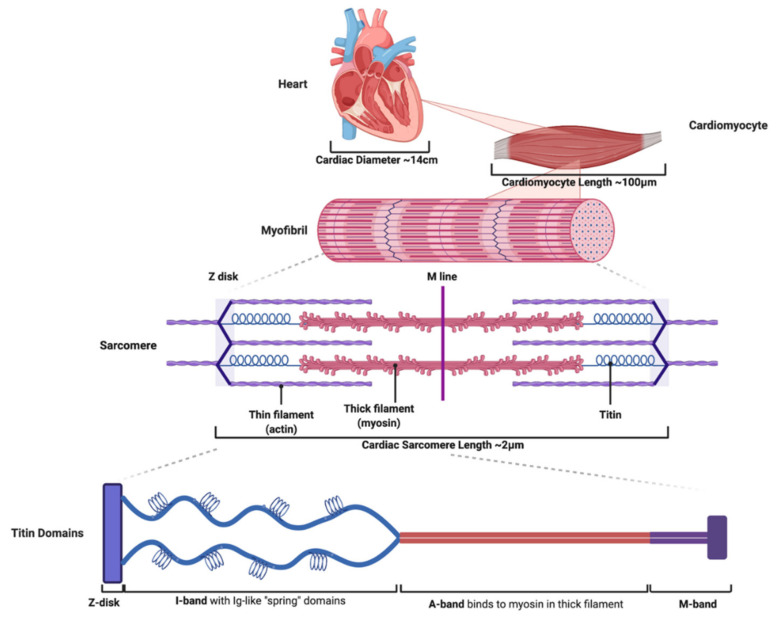
Titin structural domains serve essential functional roles as part of the sarcomere, myofibril, and cardiomyocyte. Cardiac tissue is composed of cardiomyocytes that impart contractile function. The contractile subunit of the cardiomyocyte is the myofibril, which is made of sarcomeres. Titin is an essential component of the sarcomere composed of four domains. The N-terminal domain embeds titin to the Z-disk. The I-band contains immunoglobulin-like (Ig) domains that impart extensibility and provide titin “spring-like” characteristics. The A-band binds to myosin and serves as a rigid region during contraction. The C-terminal domains embeds titin to the M-band.

**Figure 2 jcm-09-02770-f002:**
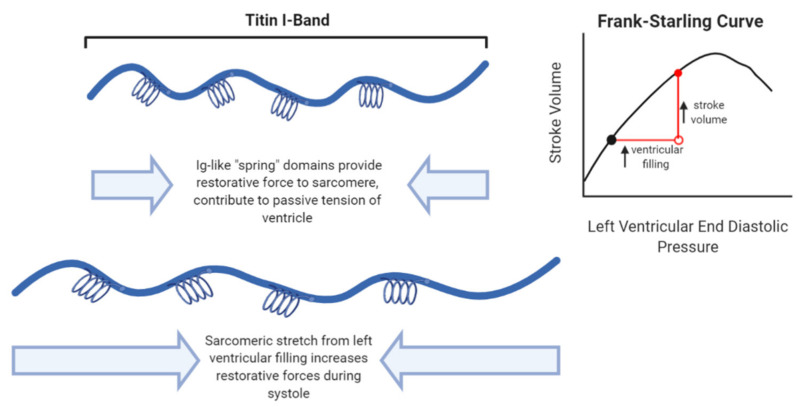
Titin’s I-band serves as a molecular spring that contributes to passive tension of the heart. The I-band is extensible due to Ig-like domains that serve as molecular springs. Because titin spans the sarcomere, the extensibility of the I-band imparts much of the resting tension of the cardiomyocyte. In addition, the I-band provides increased restorative forces when the ventricle and sarcomeres are stretched and contributes to the length dependent activation described in the Frank-Starling Curve.

**Figure 3 jcm-09-02770-f003:**
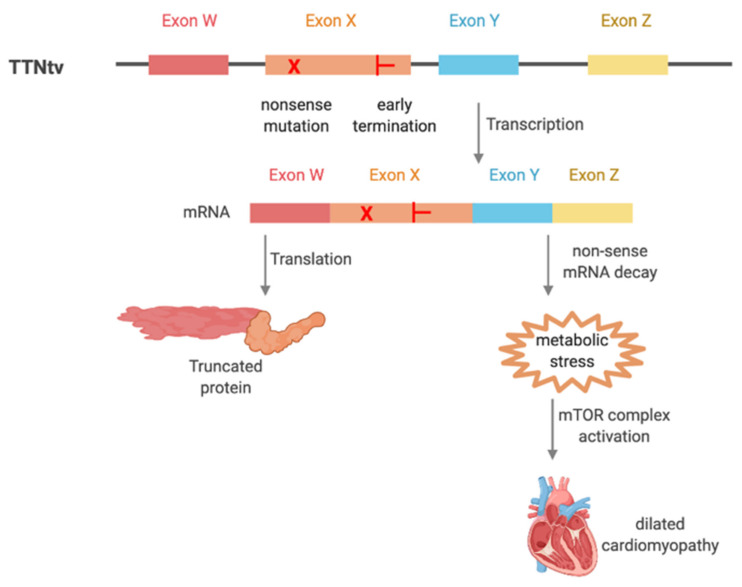
Titin truncation variations (TTNtv) lead to dilated cardiomyopathy related to haploinsufficiency and increased metabolic stress. TTNtv is likely to lead to dilated cardiomyopathy phenotype due to increased non-sense mRNA decay leading to increased metabolic stress and activation of the mammalian target of rapamycin (mTOR) complex signaling pathway. mTOR complex activation is associated with development of dilated cardiomyopathy as a downstream signaling cascade.

**Figure 4 jcm-09-02770-f004:**
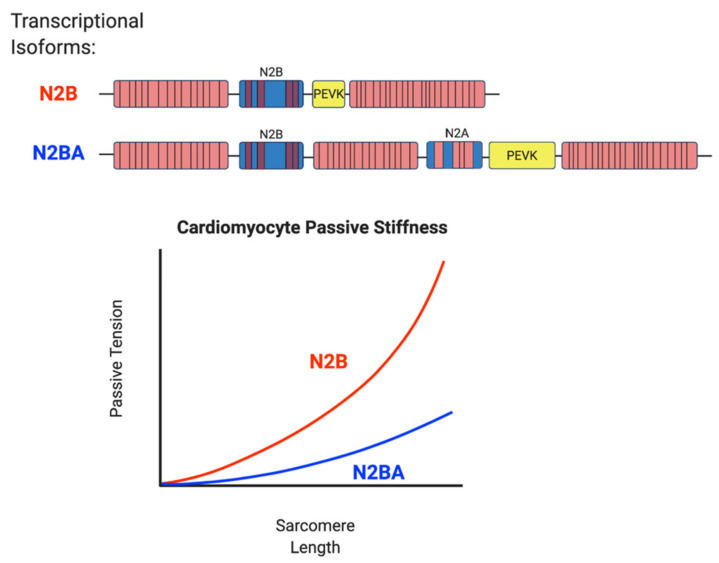
Transcriptional selection of titin isoforms affects cardiomyocyte passive stiffness. There are two major transcriptional isoforms of titin that are expressed in adult cardiac tissue. The N2B isoform has a shorter PEVK domain and has fewer Ig-like domains making it a shorter and stiffer protein. The N2BA isoform has a larger PEVK domain with more Ig-like domains and an N2A element making it a longer and softer protein. The increased size and extensibility of the N2BA isoform decreases cardiomyocyte passive stiffness and is correlated with dilated cardiomyopathy. Modified from [59].

**Figure 5 jcm-09-02770-f005:**
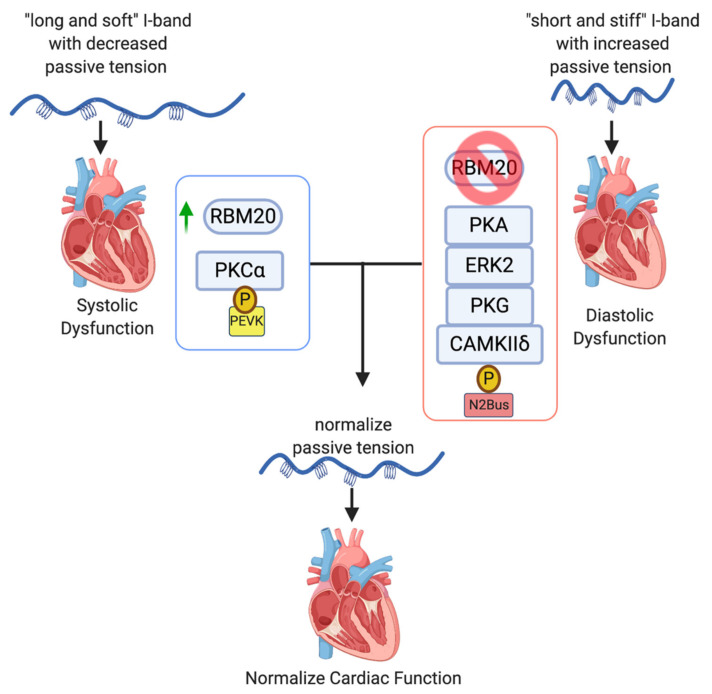
Transcriptional and post-translational modifications of titin alter passive tension and serve as therapeutic targets for treatment of heart failure. Patients with systolic dysfunction have decreased passive stiffness of the ventricle. Modifications of titin by upregulation of RNA binding motif protein 20 (RBM20) and preferencing the N2B isoform, or phosphorylation of the PEVK element can increase passive tension of the I-band and may improve cardiac function. Conversely, patients with diastolic dysfunction have increased passive tension of the ventricle. Decreased expression of RBM20 leading to increased expression of the N2BA isoform, or phosphorylation of the N2Bus element may decrease passive tension and improve cardiac function.

**Table 1 jcm-09-02770-t001:** Proposed Therapies For Treatment Of Heart Failure That Modify Titin.

Proposed Titin Modifying Therapy	Mechanism of Action	Effect on Cardiac Function
mTOR inhibitor: rapamycin	Decrease mTOR complex signaling that is activated Titin truncation variations (TTNtv) mediated mRNA decay	Improve DCM phenotype for patients with TTNtv
Antisense oligonucleotide mediated exon skipping	Bind mRNA during transcription to skip exon containing missense mutation and prevent early termination	Improve DCM phenotype for patients with TTNtv
T3 hormone, insulin	Increase RBM20 expression to transcriptionally select shorter, stiffer N2B TTN isoform	Increase passive tension to treat DCM
Cardenolides: digoxin and digitoxin	Decrease RBM20 expression to transcriptionally select longer, softer N2BA TTN isoform	Decrease passive tension to treat diastolic dysfunction
Metformin, insulin	Increase ERK2 mediated phosphorylation of N2Bus element	Decrease passive tension to treat diastolic dysfunction
Neuregulin-1 (NRG-1)	Increase ERK2 mediated phosphorylation of N2Bus element, inhibit PKC⍺ phosphorylation of PEVK element	Decrease passive tension to treat diastolic dysfunction
cGMP agonists: sildenafil, vericiguat	Increases cGMP activity to increase PKG mediated phosphorylation of N2Bus element	Decrease passive tension to treat diastolic dysfunction

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
