# Peer review of "Modifications of Titin Contribute to the Progression of Cardiomyopathy and Represent a Therapeutic Target for Treatment of Heart Failure"

_jcm, 2020, doi:10.3390/jcm9092770_

Round 1
Reviewer 1 Report
I have reviewed the manuscript 'Titin Is a Therapeutic Target for Treatment of Heart Failure: Modifications of the Giant Sarcomeric Protein Contribute to the Progression and Possibly the Reversal of Cardiomyopathy'. The manuscript is interesting and written in a good way. However, there are several minor points, which need to be revised.
1.) The title is misleading and too optimistic. Please prevent overstatements like 'possibly the reversal of cardiomyopathy'.
2.) Please explain all abbreviations like Ig (line 12) in the revised manuscript.
3.) You do not explain 'restrictive cardiomyopathy' (RCM) ? Why are you not introducing RCM in your manuscript? Even Because some TTN mutations are linked with RCM See Yael Peled, et al. Int J Cardiol 2014 Jan 15;171(1):24-30. doi: 10.1016/j.ijcard.2013.11.037.
4.) In Figure 1. Scale bars would be helpful to compare the different structures.
5.) Mutations in RBM20 are also linked with LVNC and ACM. You should also cite relevant literature for these cardiomyopathies. For a review please see: Gerull B et al., The Genetic Landscape of Cardiomyopathies, DOI: 10.1007/978-3-030-27371-2_2, In book: Genetic Causes of Cardiac Disease, Publisher: Springer Nature Switzerland AG. https://link.springer.com/chapter/10.1007/978-3-030-27371-2_2
6.) Please cite also the first manuscript describing a pathogenic TTN mutation in DCM. This landmark paper is still missing in your manuscript.
Mutations of TTN, encoding the giant muscle filament titin, cause familial dilated cardiomyopathy. Nat Genet. 2002 Feb;30(2):201-4. doi: 10.1038/ng815. Epub 2002 Jan 14.PMID: 11788824
7.) Please increase the size of Figure 3.
8.) Could you please give an overview about the different animal models for TTN and RBM20. Some of them are summarized in Genetic Animal Models for Arrhythmogenic Cardiomyopathy.
Reviewer 2 Report
The review by Tharp and colleagues aims to summarize the current evidence on titin’s contribution to normal cardiac physiology and cardiac pathophysiology. A particular emphasis is dedicated to the potential therapeutic target of this protein of the cardiac sarcomere.
The topic of the review is relevant, and its overall quality is high. The topic has been extensively described, and the nicely presented figures enrich this article making its main messages easily accessible. I think that the readers of JCM will appreciate this content.
I only found a few minor points whose improvement might increase the paper’s readability:
1) The first chapter of this review article is too general and less focused on the topic compared with the remaining text. It contains several well-known notions about cardiac physiology and pathophysiology (e.g. the cardiac cycle, the sarcomere structure, the Frank-Starling effect, or the difference between systolic and diastolic dysfunction). Perhaps it should be shortened and more focused on the physiological role of titin.
2) Similarly, in lines 55-56 the authors state: “Although titin has many complex molecular properties, its mechanical role supporting the sarcomere and providing passive tension and force modulation during contraction and relaxation has been the most well described”. However, these molecular properties are not described. Perhaps adding some detail this chapter might be more focused on the topic.
3) Line 390 please correct “treatment”
4) Figure 5 is well designed and clear. However, the addition of a table summarizing the “take-home” messages about titin as a therapeutic target might be appropriate for a clinical Journal as JCM.
